# A semi-automated method for unbiased alveolar morphometry: Validation in a bronchopulmonary dysplasia model

Thomas Salaets[1,2]*, Bieke Tack[3,4], André Gie[1], Benjamin Pavie[5], Nikhil Sindhwani[1], Julio Jimenez[6], Yannick Regin[1], Karel Allegaert[7,8], Jan Deprest[1,9,10], Jaan Toelen[1,2]

**1** Department of Development and Regeneration, KULeuven, Leuven, Belgium, **2** Department of Pediatrics, UZ Leuven, Leuven, Belgium, **3** Department of Microbiology, Immunology and Transplantation, KU Leuven, Leuven, Belgium, **4** Department of Clinical Sciences, Institute of Tropical Medicine, Antwerp, Belgium, **5** Department of Imaging and Pathology, KULeuven, Leuven, Belgium, **6** Facultad de Medicina, Universidad del Desarollo, Clínica Alemana, Santiago de Chile, Chile, **7** Department of Pharmaceutical and Pharmacological Sciences, KULeuven, Leuven, Belgium, **8** Department of Clinical Pharmacy, Erasmus MC, Rotterdam, The Netherlands, **9** Department of Obstetrics and Gynecology, UZ Leuven, Leuven, Belgium, **10** Institute for Women's Health, University College London Hospital, London, United Kingdom

* thomas.1.salaets@uzleuven.be

**Data Availability Statement:** All available data is present in the graphs as individual data points. A spreadsheet containing the individual counting results is available in the supplement.

## Abstract

Reproducible and unbiased methods to quantify alveolar structure are important for research on many lung diseases. However, manually estimating alveolar structure through stereology is time consuming and inter-observer variability is high. The objective of this work was to develop and validate a fast, reproducible and accurate (semi-)automatic alternative. A FIJI-macro was designed that automatically segments lung images to binary masks, and counts the number of test points falling on tissue and the number of intersections of the air-tissue interface with a set of test lines. Manual selection remains necessary for the recognition of non-parenchymal tissue and alveolar exudates. Volume density of alveolar septa ($V_{V_{sep}}$) and mean linear intercept of the airspaces ($Lm$) as measured by the macro were compared to theoretical values for 11 artificial test images and to manually counted values for 17 lungs slides using linear regression and Bland-Altman plots. Inter-observer agreement between 3 observers, measuring 8 lungs both manually and automatically, was assessed using intraclass correlation coefficients (ICC). $V_{V_{sep}}$ and $Lm$ measured by the macro closely approached theoretical values for artificial test images ($R^2$ of 0.9750 and 0.9573 and bias of 0.34% and 8.7%). The macro data in lungs were slightly higher for $V_{V_{sep}}$ and slightly lower for $Lm$ in comparison to manually counted values ($R^2$ of 0.8262 and 0.8288 and bias of -6.0% and 12.1%). Visually, semi-automatic segmentation was accurate. Most importantly, manually counted $V_{V_{sep}}$ and $Lm$ had only moderate to good inter-observer agreement (ICC 0.859 and 0.643), but agreements were excellent for semi-automatically counted values (ICC 0.956 and 0.900). This semi-automatic method provides accurate and highly reproducible alveolar morphometry results. Future efforts should focus on refining methods for automatic detection of non-parenchymal tissue or exudates, and for assessment of lung structure on 3D reconstructions of lungs scanned with microCT.

**Funding:** KA and TS are supported by the SAFEPEDRUG project (funded by the agency for innovation by Science and Technology in Flanders IWT SBO 130033). JT is supported by a C2 grant from KU Leuven (C24/18/101) and a research grant from the Research Foundation – Flanders (FWO G0C4419N). AG is supported by the Erasmus+ Programme of the European Commission (2013–0040). BT is holder of an FWO-SB fellowship (Research Foundation Flanders, 1153220N). YR is holder of an FWO-SB fellowship (Research Foundation - Flanders, 1S71619N). JD is funded by the Great Ormond Street Hospital Charity. The funding bodies had no role in study design, data collection and analysis, decision to publish, or preparation of the manuscript. www.safepedrug.eu www.ewi-vlaanderen.be www.fwo.be www.gosh.org www.kuleuven.be/english/research/support/if.

**Competing interests:** The authors have declared that no competing interests exist.

## Introduction

During normal in utero lung development, large saccular airway structures with thick walls, evolve into smaller thin-walled alveoli through a process called secondary septation [1]. This process increases the total amount of surface available for gas exchange, and continues into adolescence [2]. During aging, alveolar size increases and surface area decreases again by coalescence [3]. Several diseases are known to affect alveolar structure. Bronchopulmonary dysplasia (BPD) for instance is characterized by larger alveoli, with thick septa [4]. Emphysema results in larger alveoli and a decreased total alveolar surface area [5], while fibrosis locally increases septal wall thickness [6]. The alveolar structural changes are often used as primary read-outs in preclinical research on these diseases. Reproducible and unbiased methods to quantify alveolar structure are of uttermost importance for the quality of this research.

The gold standard in quantification of structure is stereology. Stereology is a set of probabilistic methods used to define the physical properties of an irregular 3D structure based on 2D transsections through the structure. It has been used in lungs to estimate volumes, areas and numbers, in an unbiased way that is free of assumptions regarding size, shape or orientation of the structures [7]. Regarding lung structure, the most commonly used read-out is the mean linear intercept of the airspaces [8], reflecting the volume to surface-ratio of the airspaces (*Lm*). Proper determination of Lm includes manual point and intersection counting of septal tissue on a high number of test points and lines [9]. Using the same septal point and intersection counts, also alveolar surface area and septal tissue or air volumes can be estimated [7].

Manual counting is very time consuming and our experience (presented in this paper) suggests important inter-observer variability. Methods to automate or facilitate counting for alveolar morphometry (sometimes compatible with a stereological approach) have been developed. STEPanizer© is a free tool that loads test systems on digital images, but still involves manual counting [10]. More recently a system for automated high–throughput analysis of whole lung slides was described, however it requires specific software [11]. Plugins or macros for freely available software to count individual fields have also been developed [12–14]. The common disadvantage of all these automated counting methods is that they do not discriminate large airways and blood vessels from lung parenchyma, and exudates or debris from septal tissue. The inability to filter out non-parenchyma or recognize inflammatory exudates leads to a significant bias of morphometric results.

The objective of this work was to develop and validate a fast, reproducible and accurate semi-automated method for counting of septal tissue points and intersections, which can be used to quantify alveolar surface area and volume, as well as mean linear intercept (*Lm*).

## Materials and methods

### Macro

A FIJI-macro [15] was designed that automatically counts the number of test points falling on reference tissue (parenchymal lung tissue, excluding conductive airways and vessels; $P_{ref}$) or septal tissue (in contrast to air; $P_{sep}$), and the number of intersections of the air-tissue interface ($I$) with a test line system. This macro automates the process of manual counting. The detection of non-reference tissue such as conductive airways, arteries, veins, pleura is still based on manual selection. In contrast, the detection of septal tissue is based on automatic thresholding, although exudates in the alveolar spaces had to be selected manually. The user has to manually define the threshold between tissue and air only in the first image of a batch (under the critical assumption that in each set of tissues are comparably processed, stained and images acquired using the same microscope settings). The macro is developed for batches of lung fields of 500x500μm, in PNG-format.

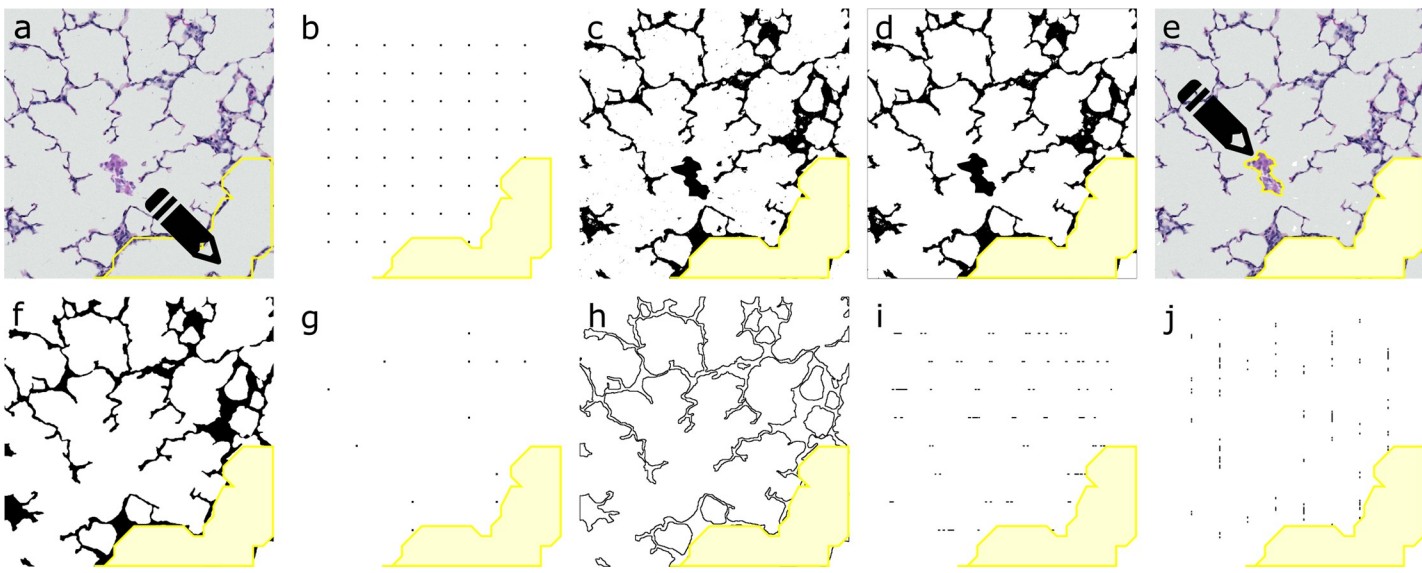

**Fig 1. Summary of image processing and analysis from lung field to $P_{ref}$, $P_{sep}$ and $I^{*}$.** (A) Non-parenchymal tissue (in this case conductive airway) is selected manually. (B) $P_{ref}$ is counted as the number of points of a 64 point test grid, outside the selection of non-parenchymal tissue. (C) Image is thresholded to binary mask. (D) Automatically all small and unconnected exudates are removed. (E) Larger and connected exudates require manual selection. (F) Mask is cleaned and smoothened. (G) $P_{sep}$ is counted as the number of points of the test grid that are common to the cleaned mask. (H) The edges of the cleaned mask are aligned. (I-J) $I$ is counted as the number of intersections of a test system of 8 horizontal and 8 vertical lines with the edge-image. $^{*}P_{ref}$ = reference points, $P_{sep}$ = septal points, $I$ = intersections.

The image processing and analysis sequence that is automatically followed for all PNG-images in a folder, is briefly described below and summarized in Fig 1.

- Resizing of the image to 680x680 pixels (pixel resolution of 0,735μm) to match the test system.

- Manual selection of the non-parenchymal tissue (conductive airways, arteries and veins) using a freehand selection tool (Fig 1A).

- Deletion of the points matching the non-parenchymal tissue on a quadratic test system of 64 points on a central counting area of 567x567μm. Counting of the remaining points in the grid: $P_{ref}$ (Fig 1B).

- Processing of the image to an 8-bit image and subsequently to a binary mask of tissue versus air by applying the user-defined threshold (Fig 1C).

- Automatical detection of exudates, as unconnected particles of 0 to 1000 pixels that do not overlap with the border of the image and removal from the mask (Fig 1D).

- Visual removal of the automatically detected exudates from a duplicated original image. Manual selection of the exudates that were not automatically detected (too large or in contact with septal tissue) and removal from the mask (Fig 1E).

- Cleaning of the mask, by filling white particles of 1 to 200 pixels within the septal tissue, by removal of outliers and by a dilate-erode sequence (Fig 1F).

- Calculation of a binary image containing all pixels that are black in both the clean mask and the test grid without the non-parenchymal points. Counting of the number of 1-pixel particles: $P_{sep}$ (Fig 1G).

- Determination of the edges of the mask and creation of an edge image (Fig 1H).

- Calculation of a binary image containing all pixels that are black in both the edge image and an image of 8 horizontal (Fig 1I) and then 8 vertical lines (Fig 1J) of 567 pixels (unit length d = 104μm or 141.75 pixels). Counting of the number of particles of both the horizontal and vertical image: *I*.

The macro generates one.csv file containing $P_{ref}$, $P_{sep}$ and *I* for every lung field. It also automatically saves intermediate images and selections for a possible visual check. The macro (S1 File, morphometry_v4.0.ijm) is available for download with this paper, together with a user manual (S2 File) and a second macro that can be used for checking the accuracy of the algorithm in batch (S3 File, checkmorphometry_v4.0.ijm).

## Artificial test images

To test and validate the counting procedure in the macro, a set of artificial binary images was made in FIJI. All images were 2040x2040 pixels and were composed of wallpaper-like repetitions of a combination of geometric figures for which surface and perimeter can be calculated. Within each square-shaped repetition the geometric figures (in white) were randomly organized against a black background. Based on the surface and perimeter of the geometric figures, a theoretical "septal volume" density ($V_{V_{sep}}$) and mean linear intercept of the geometric "airspaces" (*Lm*) could be calculated.

$$V_{V_{sep}} = \frac{2040^2 - \sum surface_{geometric\ figures}}{2040^2}$$

$$Lm = \frac{4 * (1 - V_{V_{sep}})}{\sum perimeter_{geometric\ figures}}$$

Twenty (partially overlapping) fields of 680x680 pixels were randomly extracted using the results of a random number generator as coordinates. An example is shown in Fig 2A. These 20 fields were automatically counted using a slightly modified version of the macro that omitted the steps converting a color image to a cleaned binary mask.

## Lung images

Lung slides were obtained from previous experiments in the preterm rabbit model for BPD [16, 17]. Briefly, pregnant rabbit dams underwent cesarean section on day 28 of gestation, (normal gestation is 31 days). Preterm rabbit pups were raised in an incubator with controlled temperature and humidity, in either normoxia (N; 21% $O_2$) or hyperoxia (H; 95% $O_2$). Pups were hand-fed, twice daily, by gavage, using a milk replacer appropriate for rabbits. Colostrum and probiotics were added to the feeding. Vitamin K was supplemented intramuscularly, and antibiotics were administered to prevent infections. On day 7 pups were euthanized. Experiments were approved by the Ethical Committee for Animal Research at KULeuven as P059/2016.

Lungs of preterm rabbits on day 7 were harvested "en-bloc" after thoracotomy, with the trachea and carina intact. Right bronchus was ligated and right lungs removed for other purposes. A 20G catheter was inserted and fixed in the trachea, through which the left lung was connected to a pump filled with paraformaldehyde 4%, delivering a constant hydrostatic pressure of 25cmH₂O. For 24h, the left lung was kept in a bath for immersion fixation, while connected to the system for pressure-fixation. Afterwards lungs were rinsed in phosphate buffered saline,

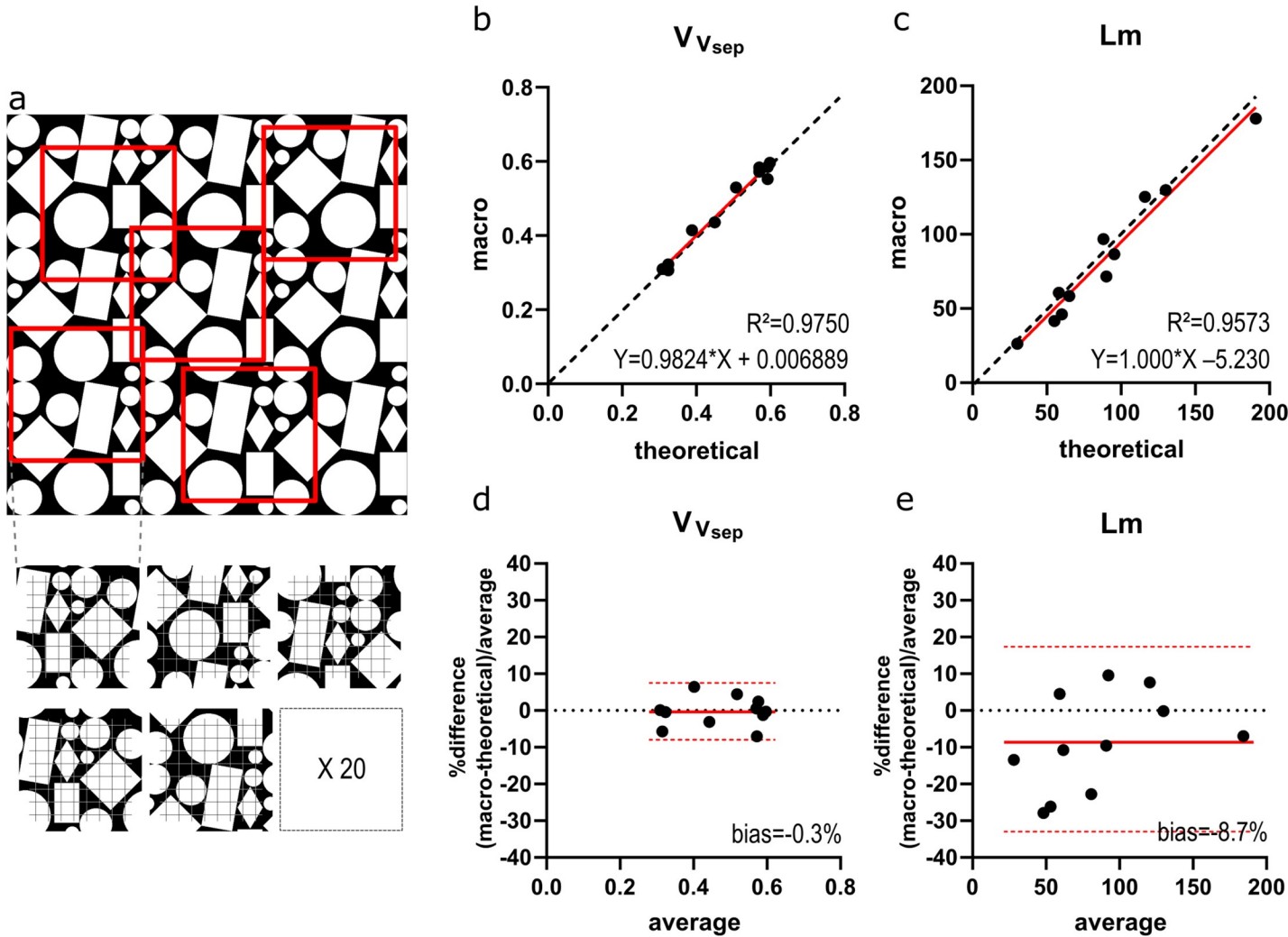

**Fig 2. Validation on artificial images.** (A) Illustration of an artificial test image with sampling strategy. (B-C) Macro results for $V_{V_{sep}}$* and $Lm$† plotted against the theoretical values. The regression line is plotted in red, dashed line represents identity (y = x). (D-E) Bland-Altman plots for $V_{V_{sep}}$* and $Lm$† plot the relative difference [(theoretical–macro)/average] to the average result. Red line indicates bias, with confidence interval (dashed red line). *volume density of alveolar septa ($V_{V_{sep}}$); †mean linear intercept of the airspaces ($Lm$).

dehydrated in ethanol and embedded in paraffin as a whole. Sagittal 5μm thick microtome sections were made and stained with hematoxylin and eosin.

Lung slides were scanned with a whole slide scanner (Axio Scan® Slide Scanner, Zen Zeiss, Oberkochen, Germany), generating 1 CZI-file for each whole lung slide. A customized Groovy Script for FIJI randomly sampled 20 fields of 500x500μm within the edges of the lung tissue, and exported them as PNG-images. The Groovy Script for random sampling can be found in S4 File (extract_some_tiles_from_tissue_czi_v02.groovy). An example of the lung sampling is shown in Fig 3A.

### Manual counting

The results of semi-automatic counting with the FIJI-macro were compared to manual counting, the current gold standard. The same 20 lung fields were converted to JPG-images for

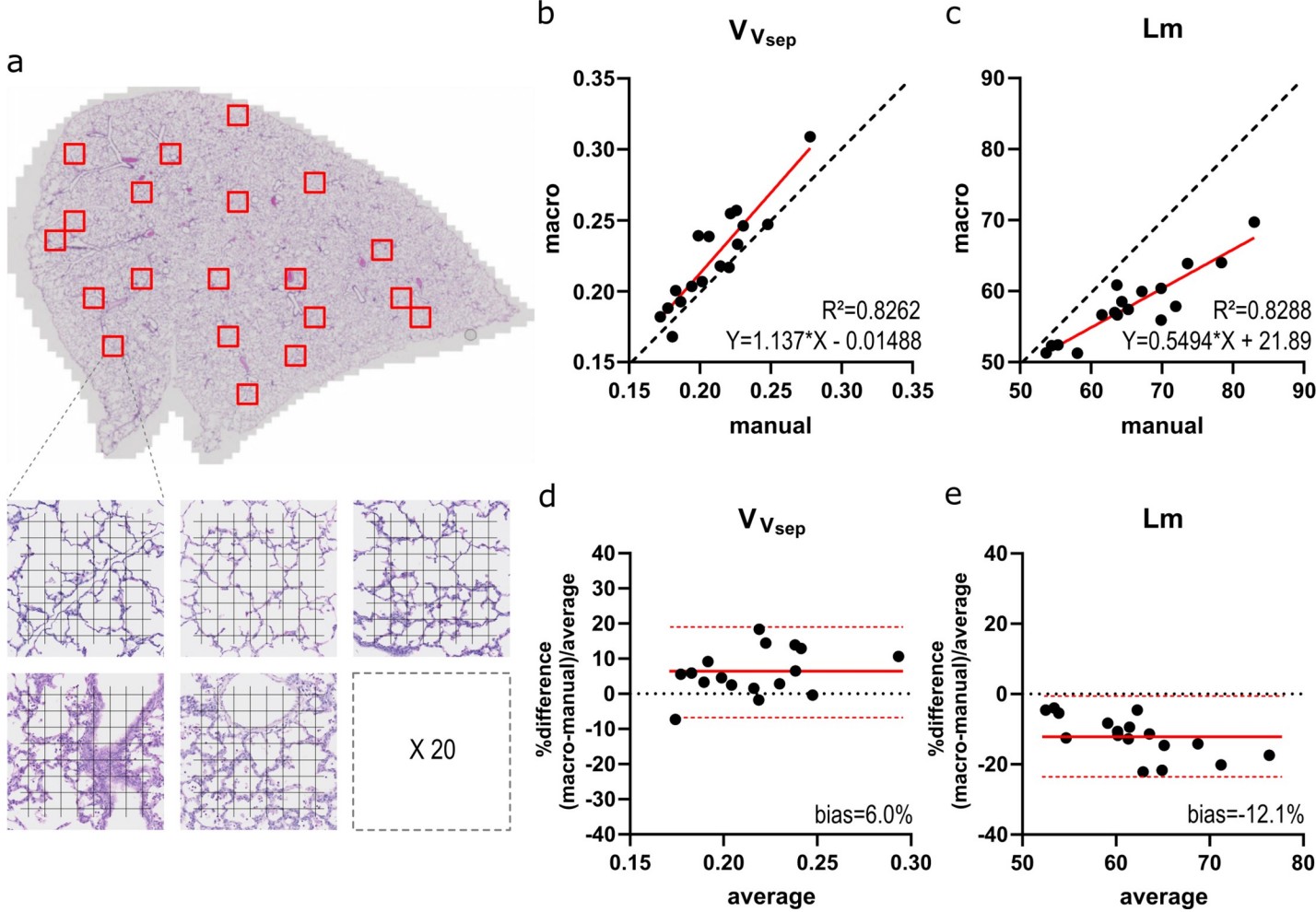

**Fig 3. Validation on lung images.** (A) Illustration of a lung slide with sampling strategy. (B-C) Macro results for $V_{V_{sep}}$ * and $Lm$† plotted against the manual values. The regression line is plotted in red, dashed line represents identity (y = x). (D-E) Bland-Altman plots for $V_{V_{sep}}$ * and $Lm$† plot the relative difference [(manual–macro)/ average] to the average result. Red line indicates bias, with confidence interval (dashed red line). *volume density of alveolar septa ($V_{V_{sep}}$); †mean linear intercept of the airspaces ($Lm$).

counting with STEPanizer [10]. A quadratic test system of 64 points and 8 horizontal and 8 vertical lines with a total length of 6670μm (d = 104 μm) was used, with an equal length and position as the test system in the macro [18]. The points falling on non-parenchymal tissue ($P_{non-par}$), $P_{sep}$ and $I$ were manually counted.

## Morphometric read-outs

Volume density of alveolar septa ($V_{V_{sep}}$) and mean linear intercept of the airspaces ($Lm$) of each test image or lung slide was calculated using the following formulas, according to the guidelines for quantitative assessment of lung structure of the American Thoracic Society [7]:

$$P_{ref} = 64 - P_{non-par}$$

$$V_{V_{sep}} = \frac{\sum P_{sep}}{\sum P_{ref}}$$

$$Lm = 2 * d * \frac{\sum P_{ref} - \sum P_{sep}}{\sum I}$$

Other read-outs such as mean transsectional wall length ($Lmw$) [19] and the surface area density of the air spaces ($S_{V_{air}}$) [7, 18] can also be calculated based on $P_{ref}$, $P_{sep}$ and $I$. When the total lung volume is available, volume and surface density can be converted to absolute volume ($V_{sep}$) and surface ($S$), respectively. For the purpose of validation of the counting macro, we only focused on $V_{V_{sep}}$ as a read-out based on point counting alone, and $Lm$ as a read-out based on both point and intersection counting.

### Validation

For validation, we first tested the macro on a set of 11 artificial test images, and compared the automatically calculated $V_{V_{sep}}$ and $Lm$ to the theoretical values. On lung images, $V_{V_{sep}}$ and $Lm$ were determined both manually and semi-automatically by a single observer for a set of fields from 17 lungs (of which 5 and 7 in plain normoxia and hyperoxia conditions respectively) and by 3 independent observers for a set of 8 lungs. The manual selection of non-parenchymal tissue and exudates in the semi-automatic macro was done independently by each of the 3 observers. Semi-automatic results of 17 lungs were compared to manual results for a single observer. The results of the 3 observers were used for assessment of inter-observer agreement (reproducibility).

Results were compared by means of linear regression and Bland-Altman plots using Graph-Pad Prism® 7.0 software (GraphPad, La Jolla, California, USA). We report $R^2$-values, a best-fit equation for the regression line and relative bias (with confidence interval) derived from the Bland-Altman plots. We calculated ICC (two way random effects model for absolute agreement) using SPSS® version 25 (IBM, Armonk, New York, USA) for inter-observer agreement between the 3 observers [20]. Results from an N and H group were compared by a t-test with Welch correction for unequal variance.

### Results

### Artificial test images

We observed a very strong correlation between the results of the macro and the theoretical results of the artificial test images. For $V_{V_{sep}}$ $R^2$ was 0.9750 and the trend line was almost on the identity line (Y = 0.9824*X + 0.006889; Fig 2B). Bias was -0.35% (CI -8.2–7.5%; Fig 2D). For $Lm$ $R^2$ was 0.9573 and the trend line was close to identity (Y = 1.000*X– 5.230; Fig 2C). The Bland-Altman plot confirmed an underestimation by the macro with a bias of -8.7% (CI -34.7–17.2%; Fig 2E).

### Lung images

For the lung images, the correlation between manual counting and the macro was less strong (Fig 3), despite the use of exactly the same fields and exactly the same test system for both manual and semi-automatic counting. $R^2$ was 0.8262 for $V_{V_{sep}}$ with semi-automatic results that somewhat overestimated the manual values (Y = 1.137*X—0.01488; Fig 3B). Overestimation was confirmed by a bias of 6.0% (CI -6.9–18.9%; Fig 3D). Almost always, counting with the macro resulted in a higher number of septal points ($P_{sep}$) in comparison to manually counting. For $Lm$ $R^2$ was 0.8288, but the trend line indicated a considerable underestimation of $Lm$

(Y = 0.5494*X + 21.89; Fig 3C) in comparison to the manual gold standard. The Bland-Altman plot in Fig 3E confirms underestimation of *Lm* with a bias of -12.1% (CI -23.5 – -0.6%). In general, overlaying the cleaned edge image over the original lung field reveals accurate automatic segmentation of the septal tissue (Fig 4).

## Reproducibility

Evaluation of inter-observer agreement of the manual results demonstrates an ICC of 0.859 (good agreement) for $V_{V_{sep}}$ and 0.643 (moderate agreement) for *Lm*. ICC increased when the semi-automatic method was used, indicating improved reproducibility. With the semi-automatic method, ICC for $V_{V_{sep}}$ was 0.956, while 0.900 for *Lm* (both excellent agreement). Pairwise comparisons between observers are shown in S1 Fig. For the manual results it becomes apparent that repetition of counting by another observer results in random error, but for Lm also in an important systematic error (over- or underestimation). Use of the macro decreases random differences overall and decreases systematic inter-observer error in the Lm-results. For $V_{V_{sep}}$ in 2/3 cases it increased systematic error, however to an arguably irrelevant extent.

## Comparison of normoxia and hyperoxia results

Both $V_{V_{sep}}$ and Lm discriminate lungs of pups raised in normoxia and hyperoxia. Manually counted $V_{V_{sep}}$ tends to increase in hyperoxia (p = 0.06) and Lm increases (p = 0.01). Use of the macro results in comparable differences between the groups: p = 0.01 and p = 0.04 for $V_{V_{sep}}$ and Lm respectively (S2 Fig).

## Discussion

The objective of this research was to develop and validate a semi-automatic counting method for lung morphometry. First, we demonstrated in a series of artificial test images that the use of the macro results in values that closely approach the true geometric characteristics of an image in general. We hypothesize that agreement is not perfect because of a potential sampling error, using a quadratic test system on a quadratic repetition of irregularly placed geometric objects. The use of the "find edges"-function, might also decrease the inner size of the geometric objects by 1 pixel at each edge and explain the slight underestimation of *Lm*. Both effects are expected to be irrelevant for completely irregular and non-binary lung images.

Second, we showed on lung samples that semi-automatic counting approached the results of manual counting, however $R^2$ was considerably lower than for the artificial test images. In comparison to manual counting it also slightly over- and underestimated $V_{V_{sep}}$ and *Lm* respectively. Despite manual counting being considered as the gold standard, it is unclear here which methods captures best the true morphometric characteristics of the tissue. A visual check of the mask reveals appropriate segmentation with the semi-automatic method, supporting the accuracy of the semi-automated method. The difference between manual and semi-automatic counting might be explained by a difference in resolution. Semi-automatic counting has a higher resolution, leading to a higher count of $P_{sep}$ and *I*. The irregularity of the alveolar surface sometimes results in 2 distinct intersections with a 1-pixel thick test line in the plugin, when the human eye in the manual method only sees the alveolar surface touching the test line once. It is plausible that the semi-automated method therefore results in higher accuracy then the manual method, however given the gold standard status of the latter this is impossible to prove.

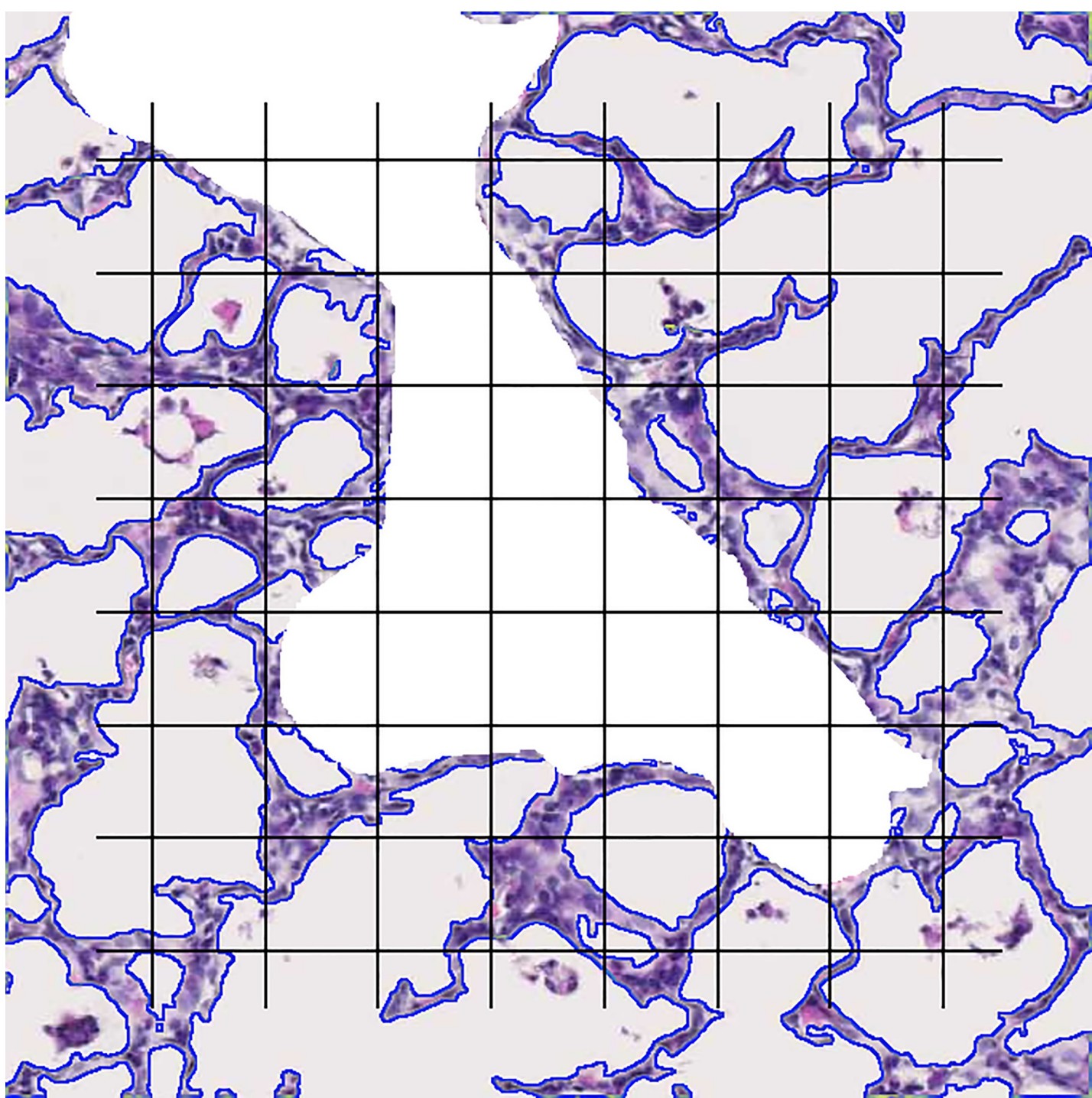

**Fig 4. Illustration of a semi-automatically segmented lung image, with overlaid test system.** Central white zone hides a manually selected vascular structure. Crossing points not falling on the vascular structure are counted as $P_{ref}$, crossing points falling on septal tissue are counted as $P_{sep}$ and intersections of the test lines with the blue line indicating the edge of the septal tissue are counted as $I$.

Third, we also observe a relevant inter-observer variability of the manual results. Semi-automatic counting on the other hand resulted in highly reproducible results. It is theoretically possible that this finding is explained by the macro making a systematic error in estimating the

true morphometric characteristics of the tissue, while manual counting results in a random error (hence its high inter-observer variability). We confirm the higher random error if manual data are compared between observers, but also note an important systematic over- or underestimation of Lm (systematic error), which was reduced by the use of the macro. We thus conclude that this macro provides a valid and more reproducible alternative to manual counting that is compatible with a stereological approach.

Additionally, we showed that counting with the macro preserves the manually counted difference in $V_{V_{sep}}$ and Lm between pups raised in hyperoxia (BPD-phenotype) and normoxia (control). Caution should however be taken to use these results as a characterization of the effect of hyperoxia in preterm rabbits. This experiment was not designed for this purpose and the included lungs do not represent a randomized group of pups from the same mothers. We however did previously use this new semi-automatic method to quantify lung structure in several experiments with complete randomization between normoxia and hyperoxia groups [21–23]. We anticipate this method can be used for research on BPD in other models, as well as on other diseases that diffusely alter lung structure (e.g. fibrosis or emphysema).

A major advantage of semi-automatic counting is efficiency. The time cost of counting however is, next to the capability of the observer to remain undistracted during this cumbersome and repetitive task, strongly dependent on the amount of inflammatory exudates and non-parenchymal areas on the fields, since these require manual selection. Previously described automatic counting methods did not exclude non-parenchymal areas or did not discriminate between septal tissue and exudates [11, 12, 14]. This can bias the results of these methods. If fields with many vascular structures or alveolar exudates (e.g lungs with important inflammation) are included, $P_{sep}$ and $I$ will probably be overestimated. If these fields are avoided, sampling will be biased towards more peripheral or less diseased areas (less inflammation), while truly random sampling is the cornerstone of a stereological approach. Further improvements in segmentation tools based on machine learning, might lead to a completely automated detection of non-parenchymal tissue or exudates. At the moment however, the human eye remains essential for these tasks.

Because it is of no relevance for the validation of the counting method, we did not focus in this paper on lung processing and sampling. We acknowledge however that it contributes to a great extent to the accuracy of morphometric read-outs in general [7]. Regardless how perfect and fast the counting method, results will be biased if processing is not optimal. Furthermore, the higher the proportion of lung that is sampled (and counted), the more closely the results approach the true structural characteristics of the tissue.

Computerized approaches offer the possibility to increase the sampling density, at a lower time cost, and thereby improve the accuracy of the result. Some authors have developed methods to quantify structural characteristics on a complete lung slide, not requiring sampling of a set of fields [11]. However, the time cost of manually cleaning these whole lung slides from non-parenchymal areas and exudates, outbalances the accuracy gained by a sample density of 100%. Furthermore, it requires specific software and hardware with a high computational speed. Finally, the advances in virtual histology should be noted. Using imaging techniques such as microCT, 3D reconstructions and segmentations have been made on which structural characteristics of pulmonary acini can be measured in 3D, instead of estimated from 2D images [24–26].

In conclusion, we developed and validated a semi-automatic method to count alveolar morphometry, compatible with a stereological approach. This semi-automatic method provides highly reproducible results in a relatively fast way. The macro is available for download as a supplement to this paper and runs on open access software (FIJI). We think the use of this

method would enhance the quality of the morphometric results in research on BPD, emphysema or fibrosis. Future efforts should go to the development of methods for automatic detection of non-parenchymal areas and exudates and for assessment of structure on 3D reconstructions of lungs scanned with microCT.

## Supporting information

**S1 Fig.**
(TIF)

**S2 Fig.**
(TIF)

**S1 File.** *Morphometry_v4.0* **plugin.** Plugin for semi-automated morphometry analysis.
(IJM)

**S2 File. User manual.** User manual of the *morphometry_v4.0* plugin.
(PDF)

**S3 File.** *Checkmorphometry_v4.0* **plugin.** Plugin to check the quality of the semi-automated segmentation (optional).
(IJM)

**S4 File.** *Extract_some_tiles_from_tissue_czi_v02* **plugin.** Plugin to randomly sample a proportion of a slide scanner image in.czi image in individual.png fields.
(GROOVY)

**S5 File. Original data.** Full results used for the analysis presented in the paper.
(XLSX)

## Acknowledgments

We acknowledge Katrien Luyten for her technical assistance with lung processing.

## Author Contributions

**Conceptualization:** Thomas Salaets, Benjamin Pavie, Nikhil Sindhwani, Julio Jimenez, Karel Allegaert, Jan Deprest, Jaan Toelen.

**Data curation:** Thomas Salaets, Bieke Tack, André Gie, Yannick Regin.

**Formal analysis:** Thomas Salaets, Bieke Tack, André Gie, Yannick Regin.

**Funding acquisition:** Jan Deprest, Jaan Toelen.

**Investigation:** Thomas Salaets, André Gie.

**Methodology:** Thomas Salaets, Bieke Tack, André Gie, Benjamin Pavie, Nikhil Sindhwani, Julio Jimenez, Jaan Toelen.

**Project administration:** Thomas Salaets.

**Resources:** Benjamin Pavie, Nikhil Sindhwani, Jan Deprest.

**Software:** Thomas Salaets, Benjamin Pavie, Nikhil Sindhwani.

**Supervision:** Benjamin Pavie, Karel Allegaert, Jan Deprest, Jaan Toelen.

**Validation:** Thomas Salaets, Bieke Tack, André Gie, Benjamin Pavie, Nikhil Sindhwani, Karel Allegaert.

**Visualization:** Thomas Salaets, Jaan Toelen.

**Writing – original draft:** Thomas Salaets, Karel Allegaert, Jaan Toelen.

**Writing – review & editing:** Thomas Salaets, Bieke Tack, André Gie, Benjamin Pavie, Nikhil Sindhwani, Julio Jimenez, Yannick Regin, Karel Allegaert, Jan Deprest, Jaan Toelen.

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
