## [Decision Letter · Decision Letter 0]

23 Jul 2020

PONE-D-20-17619

A semi-automated method for unbiased alveolar morphometry: validation in a bronchopulmonary dysplasia model

PLOS ONE

Dear Dr. Salaets,

Thank you for submitting your manuscript to PLOS ONE. After careful consideration, we feel that it has merit but does not fully meet PLOS ONE’s publication criteria as it currently stands. Therefore, we invite you to submit a revised version of the manuscript that addresses the points raised during the review process.

Please note that the original Reviewer #2 declined to review the manuscript after initially agreeing to do so.  To avoid further delay, I reviewed the manuscript myself, so I am Reviewer #2.  Along with my critique, I think it is particularly important to demonstrate that the macro can distinguish V-Vsep and Lm values from hyperoxic and normoxic lung sections. A more thorough discussion of how the macro calculated values vary from literature values is warranted. Also, both Reviewer #1 and I would like to see a more detailed analysis and discussion of inter-observer variability in manual morphometric analysis.

We look forward to receiving your revised manuscript.

Kind regards,

Michael Koval

Academic Editor

PLOS ONE

Journal Requirements:

Reviewers' comments:

Reviewer's Responses to Questions

**Comments to the Author**

1. Is the manuscript technically sound, and do the data support the conclusions?

Reviewer #1: Yes

Reviewer #2: Partly

2. Has the statistical analysis been performed appropriately and rigorously? 

Reviewer #1: Yes

Reviewer #2: Yes

3. Have the authors made all data underlying the findings in their manuscript fully available?

Reviewer #1: Yes

Reviewer #2: Yes

4. Is the manuscript presented in an intelligible fashion and written in standard English?

Reviewer #1: Yes

Reviewer #2: Yes

5. Review Comments to the Author

Reviewer #1: This well-written manuscript describes a semi-automated method for determining lung morphometry that meets a critical need in the field of development lung biology and in the study of lung disease. The authors clearly demonstrate the variability between using the more common manual counting methods to determine mean linear intercept and septal thickness. They have made their app available for download in the supplement and provided a user manual that is easy to follow. While this is not the first automated counting method to be described, much of the other software and plugins are proprietary, and this free plug-in works with the ImageJ, which is also freely available to all scientists.

Minor: One issue with the methodology is that manual counting is currently the "gold standard" for validation even though manual counting is not as accurate for all of the reasons described by the authors in their introduction. While the authors do address this issue in the text, perhaps a more detailed discussion on this point would be helpful.

Reviewer #2: In this manuscript the authors present a new semi-automated method for measuring lung morphology. There are several advantages to this method: it is public domain, easy to use and performs well against an artificially generated image. It is also very effective in outlining septal borders (Fig 4). However when applied to actual lung slices, there is a systemic bias in the measurements when compared with manual quantitation. This is especially a concern with mean linear intercept (Lm) which was consistently underestimated when compared with values obtained by manual counting done by three independent observers. The study does not compare calculations between BPD (hyperoxic) and control (normoxic) premature rabbit lungs, either as generated by the authors or with values from the literature. This combined with the systematic overestimation of V-Vsep and underestimation of Lm raises some concerns that need to be addressed.

Specific comments:

1. Lines 135-141: It looks as though there were lung sections from both hyperoxic and normoxic pups yet the data is aggregated so it is difficult to tell whether both types of sections were included in the analysis. It is especially critical to show whether the macro calculates a difference in V-Vsep and Lm between BPD and non-BPD lungs.

2. In reference 16, the authors present Lm values of ~70 (normoxia) and ~95 (hyperoxia) in a comparable BPD rabbit model. How do these compare with the values obtained manually and using the macro in the current study? Can the macro be applied to images from the literature and compared with literature values for V-Vsep and Lm as an added validation method?

3. Although the manual morphometry is presented in the supplement, including a more detailed analysis of systematic bias between the observers in the main body of the manuscript would be interesting. (listed as not shown on line 240)

4. Figure 3d,e. It looks as though the graph in 3d has a bias of -12.1 and 3e has a bias of +6.0. Are these figures reversed?

5. Lines 83-85. "The user has to manually define the threshold between tissue and air only in the first image of a batch (under the critical assumption that all images in a batch are stained in the same batch and acquired with the same microscope settings)." This is a bit confusing. Consider changing to: "The user has to manually define the threshold between tissue and air only in the first image of a batch (under the critical assumption that in each set of tissues are comparably processed, stained and images acquired using the same microscope settings)."

6. PLOS authors have the option to publish the peer review history of their article (what does this mean?). If published, this will include your full peer review and any attached files.

Reviewer #1: No

Reviewer #2: No

---

## [Author Response · Author response to Decision Letter 0]

6 Sep 2020

Rebuttal letter

We thank the editor for considering our manuscript for publication in PlosOne. Below we respond point-by-point (in italics) to the questions raised by the reviewers. 

Reviewer #1: This well-written manuscript describes a semi-automated method for determining lung morphometry that meets a critical need in the field of development lung biology and in the study of lung disease. The authors clearly demonstrate the variability between using the more common manual counting methods to determine mean linear intercept and septal thickness. They have made their app available for download in the supplement and provided a user manual that is easy to follow. While this is not the first automated counting method to be described, much of the other software and plugins are proprietary, and this free plug-in works with the ImageJ, which is also freely available to all scientists.

Minor: One issue with the methodology is that manual counting is currently the "gold standard" for validation even though manual counting is not as accurate for all of the reasons described by the authors in their introduction. While the authors do address this issue in the text, perhaps a more detailed discussion on this point would be helpful.

We included a pairwise comparison of manual counting results, to discuss in more detail the problem with manual counting: a random error, but more importantly for Lm also a systematic error. 

Reviewer #2: In this manuscript the authors present a new semi-automated method for measuring lung morphology. There are several advantages to this method: it is public domain, easy to use and performs well against an artificially generated image. It is also very effective in outlining septal borders (Fig 4). However when applied to actual lung slices, there is a systemic bias in the measurements when compared with manual quantitation. This is especially a concern with mean linear intercept (Lm) which was consistently underestimated when compared with values obtained by manual counting done by three independent observers. The study does not compare calculations between BPD (hyperoxic) and control (normoxic) premature rabbit lungs, either as generated by the authors or with values from the literature. This combined with the systematic overestimation of V-Vsep and underestimation of Lm raises some concerns that need to be addressed.

Specific comments:

1. Lines 135-141: It looks as though there were lung sections from both hyperoxic and normoxic pups yet the data is aggregated so it is difficult to tell whether both types of sections were included in the analysis. It is especially critical to show whether the macro calculates a difference in V-Vsep and Lm between BPD and non-BPD lungs.

For this validation of methodology we indeed aggregated data from different groups raised in normoxia and hyperoxia. We intended to validate the use of this macro regardless of its value to discriminate normoxia and hyperoxia pups. This discriminative capacity is indeed useful for our research line, but showing that there is a difference between normoxia and hyperoxia does not prove that the macro generates results that are close to the true morphometric characteristics of the lung. 

Upon request of the reviewer we added a comparison of normoxia and hyperoxia samples. Of the 17 lungs used for manual to macro comparison, 5 were plain normoxia and 7 were plain hyperoxia. The other 5 pups were in comparable conditions, however received a slightly different experimental treatment. We show that there are comparable differences between the groups whether manual counting or the macro was used to obtain the results. We however think these results should be interpreted with caution. In our therapeutic experiments we always ensure proper randomization of pups from the same mother. For this validation study several samples from different mothers were taken together and no proper randomization has happened between the pups of the included samples. 

2. In reference 16, the authors present Lm values of ~70 (normoxia) and ~95 (hyperoxia) in a comparable BPD rabbit model. How do these compare with the values obtained manually and using the macro in the current study? Can the macro be applied to images from the literature and compared with literature values for V-Vsep and Lm as an added validation method?

See answer above. 

If one would compare reported values, over time a trend towards less sick animals would be observed. By increasing our experience with the model and improving our care techniques we noticed that animals got less sick. Furthermore the counting method has changed tremendously over years. This manuscript finalizes the many quality improvements we have implemented to our histology workflow. Direct comparison to historical data will therefore be impossible. 

Nevertheless we have in the meanwhile published several studies using this algorithm in the model, showing that it can be used to discriminate normoxia, hyperoxia and treated samples (1, 2). Not only Lm or Vvsep but also derived measures have been used. 

3. Although the manual morphometry is presented in the supplement, including a more detailed analysis of systematic bias between the observers in the main body of the manuscript would be interesting. (listed as not shown on line 240)

We included a pairwise comparison of manual counting results, to discuss in more detail the problem with manual counting: a large random error, but more importantly for Lm also a systematic error. 

4. Figure 3d,e. It looks as though the graph in 3d has a bias of -12.1 and 3e has a bias of +6.0. Are these figures reversed?

We thank the reviewer for remarking this error. After careful revision we also noted some inconsistencies in the calculation of bias (+ or -) in both figures 2 and 3. Figures and the describing text in the result section have been corrected. 

5. Lines 83-85. "The user has to manually define the threshold between tissue and air only in the first image of a batch (under the critical assumption that all images in a batch are stained in the same batch and acquired with the same microscope settings)." This is a bit confusing. Consider changing to: "The user has to manually define the threshold between tissue and air only in the first image of a batch (under the critical assumption that in each set of tissues are comparably processed, stained and images acquired using the same microscope settings)."

Has been changed. 

1. Salaets T, Aertgeerts M, Gie A, Vignero J, de Winter D, Regin Y, et al. Preterm birth impairs postnatal lung development in the neonatal rabbit model. Respir Res. 2020;21(1):59.

2. Salaets T, Gie A, Jimenez J, Aertgeerts M, Gheysens O, Vande Velde G, et al. Local pulmonary drug delivery in the preterm rabbit: feasibility and efficacy of daily intratracheal injections. Am J Physiol Lung Cell Mol Physiol. 2019;316(4):L589-l97.

---

## [Editor Report · Decision Letter 1]

9 Sep 2020

A semi-automated method for unbiased alveolar morphometry: validation in a bronchopulmonary dysplasia model

PONE-D-20-17619R1

Dear Dr. Salaets,

We’re pleased to inform you that your manuscript has been judged scientifically suitable for publication and will be formally accepted for publication once it meets all outstanding technical requirements.

Kind regards,

Michael Koval

Academic Editor

PLOS ONE

---

## [Editor Report · Acceptance letter]

14 Sep 2020

PONE-D-20-17619R1 

A semi-automated method for unbiased alveolar morphometry: validation in a bronchopulmonary dysplasia model 

Dear Dr. Salaets:

I'm pleased to inform you that your manuscript has been deemed suitable for publication in PLOS ONE. Congratulations! Your manuscript is now with our production department. 

Kind regards, 

on behalf of

Dr. Michael Koval 

Academic Editor

PLOS ONE